# Injection and sexual risk among people who use or inject drugs in Kampala, Uganda: An exploratory qualitative study

**Julia Dickson-Gomez**[1☯¤]*****, **Wamala Twaibu**[2], **Erica Christenson**[3☯], **Katende Dan**[2], **Ronald Anguzu**[1], **Ethan Homedi**[1], **Nazarius Mbona Tumwesigye**[4]

**1** Division of Epidemiology, Institute for Health and Equity, Medical College of Wisconsin, Milwaukee, Wisconsin, United States of America, **2** Uganda Harm Reduction Network, Kampala, Uganda, **3** Center for AIDS Intervention Research, Medical College of Wisconsin, Milwaukee, Wisconsin, United States of America, **4** Makerere University, School of Public Health, Kampala, Uganda

☯ These authors contributed equally to this work.
¤ Current address: Division of Epidemiology, Institute for Health and Equity, Medical College of Wisconsin, Milwaukee, Wisconsin, United States of America
* jdickson@mcw.edu

**Data Availability Statement:** Data included in this paper include excerpts from in-depth interviews with 30 illicit drug users from Kampala, Uganda.

## Abstract

### Background

Countries in Sub-Saharan Africa (SSA) have seen rapid increases in injection drug use since 2008. In Uganda, the Global Sate of Harm report and studies conducted by Makerere University Crane Surveys have estimated HIV prevalence among people who inject drugs (PWID) at approximately 17%. The objective of the research was to document injection and other drug-related risks among people who use drugs in Uganda to develop comprehensive HIV/HCV prevention interventions.

### Methods

Between August and September 2018, we conducted qualitative interviews among male and female people who use drugs. Interview topics included the availability and accessibility of clean syringes, injection risks, overdose, sexual-risk behaviors, and the availability and accessibility of harm reduction and drug treatment services.

### Results

Participants reported several injection-related risks including sharing and reusing syringes, pooling and mixing drugs in the same container, measuring drugs using syringes, getting prefilled injections from dealers, being injected by other people who inject drugs, and using contaminated water or blood to dilute drugs. Participants reported a scarcity of harm reduction services, although a few appear to have participated in the syringe exchange pilot conducted by the Uganda Harm Reduction Network (UHRN). Even fewer reported knowing organizations that helped people who use drugs abstain from or reduce their use. Medication assisted therapy (MAT) and naloxone to reverse overdoses are not currently available.

The data cannot be shared even upon request because our consent forms stated that data would not be shared with anyone outside the research team. The Institutional Review Boards at Makerere University School of Public Health and the Medical College of Wisconsin has thus restricted access to the data set. Further questions can be directed to Kathryn Gaudreau at the Medical College of Wisconsin IRB, kgaudrea@mcw.edu, 414-955-7774 and Lynn Atuyande, at Makerere University, wtusiime@musph.ac.ug (256) 393-291-397.

**Funding:** EH received a fellowship from the Elaine Kohler Summer Research Fellowship in Global Health. The funders had no role in the study design, data collection and analysis, decision to publish, or preparation of the manuscript.

**Competing interests:** The authors have declared no competing interests exist.

## Conclusions

Comprehensive prevention and treatment services are needed in Uganda and should include expanded syringe exchange programs, social network HIV testing, HCV testing, provision of naloxone and MAT, and linkage to and retention in HIV care.

## Introduction

HIV prevalence in Uganda is among the highest in the world, with national prevalence estimated at 6.2% [1] and prevalence among high risk populations such as fisherfolk and female sex workers at 30% [2] and 33% [3] respectively. While Uganda has been the site of abundant HIV prevention and research efforts, it is one of only two countries in Sub-Saharan Africa (SSA) in which HIV incidence is increasing, particularly in the general population [4]. There is some indication that unaddressed factors including injection and non-injection drug use may be impairing Uganda's HIV prevention efforts [5, 6]. Countries in SSA have seen rapid increases in injection drug use since 2008 including heroin and cocaine [7–10], and these drugs are also present and increasingly used in Uganda [11–13]. Heroin and cocaine injection are highly prevalent in the coastal areas of Tanzania and Kenya which have long acted as trans-shipment points but have increasingly opened local markets. A recent review of studies found a substantial proportion of heroin users who inject in Kenya and Tanzania [14] and that these groups may have high HIV prevalence rates. In Kenya, of 101 current injectors, 52.5% were found to be HIV positive, attributed mainly to sharing injection equipment compared to 13.5% of heroin users who had not injected [14]. In Uganda, the Global Sate of Harm report and studies conducted by Makerere University Crane Surveys have estimated HIV prevalence among people who inject drugs (PWID) at approximately 17% [1, 11]. In 2018, the Kampala Place Study estimated over 18,000 people who inject drugs (PWID) in Uganda [15]. A high number of women who use injection and non-injection drugs report selling sex (79%) and may have significantly higher rates of HIV infection [16]. There is no reliable data on the prevalence of HCV as PWID are not tested for this in Uganda or other SSA countries [12]

Despite the effectiveness of existing HIV prevention efforts for people who inject drugs, such efforts are inadequate or non-existent in many SSA countries [14, 17]. Uganda has recently piloted a syringe exchange program [18] and has pledged to devote more resources to prevent HIV among PWID including providing medication assisted therapy (MAT) such as buprenorphine to help drug users abstain from more harmful opioids like street heroin, or injection risks; the President's Emergency Plan for AIDS Relief (PEPFAR) and the Global Fund key population investment fund have pledged money and resources to begin MAT by this year in three centers in Kampala. Tanzania and Kenya recently opened methadone clinics in 2016 and 2018, respectively [19]. These efforts are likely insufficient to combat HIV in PWID in East Africa and have been hampered by addictophobia on the part of providers, stigma of people living with HIV (PLH) and PWID and a lack of political will [20]. Most researchers argue that a combination of approaches are needed including provision of clean syringes through syringe exchange programs (SEPs), frequent HIV testing and linkage to HIV care, antiretroviral therapy initiation after infection to reduce risk of onward transmission through viral suppression, medication assisted therapy, psychosocial support and pre- and post-exposure prophylaxis [20–22]. In addition, modelling projections suggest that very high coverage of ART, SEPs, and MAT in combination are necessary to reduce HIV incidence of PWID by more than 50%; very high intensity and coverage of single interventions is necessary

to achieve similar effects, otherwise single interventions are unlikely to be effective [21]. Very few countries provide all these harm reduction services and services are often fragmented and rarely offered in one place. For example, many studies have shown that the integration of substance abuse treatment in HIV care can improve adherence to ART, but such programs are still rare [23–25]. At present, only HIV testing and linkage to care is consistently provided to drug users and only in Kampala through the Uganda Harm Reduction Network (UHRN) outreach to drug hotspots and peer navigation to government HIV clinics in which ART is provided free of charge. However, efforts are underway to increase harm reduction centers to the rest of the country that include MAT, syringe exchange and PrEP.

The current study presents results from qualitative interviews with 30 people in Kampala, Uganda who use injection and non-injection drugs to determine their behavioral risks for HIV and the availability and accessibility of resources to reduce the harms associated with injection such as clean syringes, non-pharmaceutical drug treatment such as counseling or 12-step programs (as MAT was not available at the time of study), condoms and HIV testing.

## Methods

We conducted 30 in-depth interviews with people who used illicit drugs from drug-use hotspots in Kampala, Uganda. Interviews assessed the physical and social context in which drugs were consumed that increased or decreased injection- and drug-related sexual risk. Interviews also explored participants' access to syringes, experiences with HIV prevention programs, HIV testing services, HIV treatment, and any substance abuse prevention or treatment services. This study was approved by Institutional Review Boards at the Medical College of Wisconsin, Makerere University School of Public Health, and the Uganda National Council for Science and Technology.

### Eligibility criteria and interview sampling characteristics

Eligibility requirements included being 18 or older, having used an illicit drug in the last 30 days, and being able to provide informed consent in English or Luganda. Drug use was kept open in order to explore the types of drugs most commonly used. We include both people who inject drugs (PWID, n = 17) and people who use drugs (PWUD) but do not inject (n = 14), because while injection with contaminated syringes is one of the most efficient ways to become infected with HIV, PWID and PWUD can be at risk for becoming infected through high-risk sexual behaviors. In addition, PWID and PWUD often are parts of the same drug using networks. We purposively oversampled to include more women than in the general drug using population.

### Recruitment and consent

We used a combination of standard outreach techniques and participant referral to identify and recruit eligible participants for in-depth interviews. Members of the Uganda Harm Reduction Network who served as Research Assistants to the project recruited directly from selling "hotspots" in Uganda. Screening questions verified that the potential participant used cocaine, heroin, methamphetamine, or methaqualone in the past 30 days. We also asked participants about their alcohol use and sexual activity to disguise eligibility criteria to avoid false reporting of eligibility. Participants under 18 years old were excluded from in-depth interviews due to potential risks involved in obtaining parental informed consent for minors to participate in interviews focusing on illicit drug use. Eligible participants, or participants who were ineligible but indicated that they had previously used drugs, were asked to refer someone they know who uses drugs to the study. Again, we asked seed participants to bring or send potential participants to field staff for screening, and staff never directly approached an individual who was

identified by a participant. Luganda informed consent forms were read aloud to participants by research interviewers. After reading the consent form, potential participants were asked if they had any questions and were asked questions by the interviewer to test their understanding of the project and their decision to participate or not. Participants who agreed to participate were asked to sign the form or to make their mark if illiterate to signify their consent. The research interviewer witnessed consent and signed the consent form.

### In-depth interview procedures and content

Interviews took place in a private location in the community, for example a private room in a community center or the project car. We informed participants that all the information they gave us was confidential. Interviews took about 1½ hours to complete and were tape-recorded, except for interviews with three participants when a tape recorder was not available. Detailed notes were taken of these interviews. Interviews were conducted by an outreach worker with experience in qualitative and quantitative research at the UHRN, and a graduate student who is a native Ugandan. Interviews were conducted in Luganda. All interviews were transcribed and translated into English by a bilingual translator.

Interviews contained both a structured quantitative survey as well as open-ended qualitative questions. The quantitative portion included demographics: age, employment, income level, sources of income, and education level. We also assessed the quantity and frequency of use of various drugs. This brief demographic interview was followed by an open-ended ethnographic interview in which the following topics were explored among others: 1) drug use history; 2) subsistence strategies to support drug use; 3) where and how injection equipment is obtained, and a description of injection practices including how drugs are prepared or shared; 4) sexual experiences and HIV risk behaviors while using drugs; 5) experiences trying to quit or cut down on drugs including knowledge of places that offer drug treatment or people who have successfully quit; and 6) any experiences with harm reduction services such as SEPs and HIV testing.

### Qualitative analysis

All taped interviews were transcribed, and text data was coded and analyzed for key themes and patterns of response using MAXQDA software. All authors participated in developing the code book. Authors 3, a US-based MA qualitative researcher, 5, a graduate student in Public Health from Uganda and 6, a US medical student coded data while author 3 checked coding for consistency and accuracy. Disagreements were discussed and resolved by consensus. Data analysis was primarily descriptive and included drug and sexual risks participants described, contextual reasons for these risks, and the availability and accessibility of harm reduction services to reduce risks. New themes that were iteratively discovered during analysis were added to the final coding tree. The coding tree is shown in Table 1 below.

## Results

Seventeen of the 30 participants reported injection drug use (57%). Of these, 65% reported injecting cocaine, 52% heroin, and 35% injected both heroin and cocaine. Participant characteristics are described in Table 2 below.

### Injection risk behavior

Participants reported buying syringes from pharmacies, sharing them from friends or obtaining them from drug dealers. Only three reported obtaining syringes from a syringe exchange (see below). Nearly all participants who injected drugs reported sharing injection equipment.

**Table 1. Coding tree.**

| |
| --- |
| Injection |
| Types of drugs |
| Marijuana |
| alcohol |
| heroin |
| cocaine |
| other |
| Police |
| Safety/ avoiding police and detection |
| Butabika hospital |
| Prison |
| Drug confiscation |
| Torture |
| Quit attempts |
| Withdrawal |
| Overdose |
| Frequency & quantity of use |
| Initiation of drug use |
| Slums |
| Ghettos |
| Drug use networks |
| Subsistence activities |
| drug selling/ flipping/ transporting (by users) |
| robbery/ crime |
| trash pickup/ recycling |
| Employment |
| Activities when using |
| Sex work/ exchange/ paying for sex |
| Availability of syringes |
| friends |
| dealers |
| pharmacy |
| syringe exchange/ satellite distribution |
| Injection risks |
| sharing |
| reusing |
| flushing |
| teaching |
| mixing/ measuring |
| cleaning |
| Prevention outreach |
| Credit/ reciprocity among users |
| School/ education |
| Hiding/ concealing drug use |
| HIV awareness |
| peer conversations |
| status |
| stigma |

(*Continued*)

**Table 1.** (Continued)

| |
| --- |
| sex risk (including knowledge) |
| testing |
| Condoms |
| Brothels |
| Substance use treatment/ lack of |
| Substance use/ drug stigma |
| Cost of drugs |
| Homelessness/ Housing |
| Drug Selling |
| Delivery |
| Organization |
| Rules |
| Sites |
| Drug Using |
| Sites |
| Site rules |

While syringes are legally available for sale in Uganda, most participants who injected reported that they were too expensive to buy for a single use, or even for a single person. As a result, people who injected drugs often shared with each other, like the man below who reported injecting cocaine daily.

Interviewer: Under which circumstances do you share those syringes?

Participant 4: You see, the way those syringes are expensive, you have to share even needles as well as alcohol. We have to share, today it's me and tomorrow it might be him without any money on him. (23-year-old man)

In this situation of scarcity, even people who knew the risks of sharing and took precautions to protect themselves reported that they would share their used syringes with other users. This was sometimes done voluntarily to help others avoid withdrawal while at other times, people reported searching for discarded syringes when they did not have one. The woman below injected 80 times a day but insisted she would only sniff if she did not have a syringe.

**Table 2. Participant characteristics.**

| | Men, n (%) | Women, n (%) | Total, n (%) |
| --- | --- | --- | --- |
| **Sample Characteristics** | **n = 20** | **n = 10** | **n = 30** |
| Age (mean, SD) | 27.1 (4.3) | 30 (5.3) | 28.1 (4.7) |
| Injection Drug Use | 9 (45) | 8 (80) | 17 (57) |
| Drugs injected | | | |
| Just Heroin | 0 | 3 (30) | 3 (10) |
| Just Cocaine | 4 (20) | 2 (20) | 6 (20) |
| Both | 5 (25) | 3 (30) | 8 (27) |
| HIV | | | |
| Had a test | 16 (80) | 8 (80) | 24 (80) |
| HIV+ | 2 (10) | 3 (30) | 5 (17) |
| Sex work | 0 | 7 (70) | 7 (23) |
| Exchanged sex (excluding sex workers) | 5 (25) | 3 (30) | 8 (27) |

Participant 1: Syringes sometimes get lost and some use them in a group. Some syringes are shared in a group of two to three people. But that is harmful, dangerous because of transmission of HIV/AIDS. That's why when I don't have my own, I don't use. Others use and give others to use. You know am educated at least but others share the syringe. When I don't have the syringe, I don't use the injection. When I use a syringe and throw it away, they search for it and sometimes we fight for it.

Interviewer: So, when you give them, they share the needle?

Participant 1: They ask me, and I give them, but I don't use it again. So, we need more syringes to reduce on the rate of HIV infection. (20-year-old woman)

Syringes can also be bought from people on the street, not directly from pharmacies. However, many participants reported that people who sell on the streets often purchase all available syringes from the local pharmacy, creating a monopoly and selling syringes even more expensively.

Interviewer: Can you buy [syringes] somewhere else?

Participant 1: Yes, they are there, they sell but expensively. That's why we use in groups and it's dangerous. They are big people who come and buy all leaving us with an option but sharing the syringes. (20-year-old woman)

The expense of syringes also led many to reuse syringes many times. This sometimes led to syringe clogging and contributed to abscesses among injectors.

Interviewer: So, for you to know that this syringe is no longer necessary for usage and you throw it away, how many times do you use it?

Participant 12: Why throw it away?

Interviewer: You don't throw away?

Participant 12: They read the budget, where were you? All prices were hiked including that of syringes, so we don't easily throw them away. (30-year-old man, reported injecting cocaine every 30 minutes every day)

Syringes were generally replaced when they no longer functioned, in other words, clogged or stuck.

Interviewer: So, when do you replace the needle?

Participant 1: Sometimes you push, and it's stuck and not working. That's when you stop. Others use one needle for a week. But it is also not good. Every time you inject you need to buy needles-syringes. Even if you want to use the needle like three times a day then you need to buy three needles. (20-year-old woman, injects 80 times a month)

Syringes were also shared because people who inject drugs often pooled their money to buy drugs and, as a result, mixed the drugs in the same container and drew the drugs from it, or injected a portion from the same syringe. While PWIDs recognized that this was risky, it also bonded them together in their shared risk.

Interviewer: Have you ever shared needles with anyone else? In what circumstances have you done this?

Participant 22: Yes, especially when we did not have enough money to have enough drug. Sharing is a sign of brotherhood in this location. That is why we call each other "blood". (29-year-old man, injects heroin and cocaine every day)

Syringes were also shared when PWID bought premeasured syringes from drug dealers. PWID did not know whether these were clean or used syringes and reported returning syringes to dealers after injecting.

Interviewer: Tell me where you obtain needles or syringes to inject your drugs?

Participant 28: I obtained needle and syringe from a drug seller. He gave me already mixed drug. I did not take part in mixing the drug. And after him injecting me, the used ones were taken by him. (27-year-old woman, used to but no longer injects)

Another risk associated with injection in Uganda that is perhaps not as prevalent in developed countries, is the availability and accessibility of clean water to dilute drugs for injection. Very few participants reported buying bottled water or receiving saline water for injection. Most often, participants reported using tap water to mix drugs.

Interviewer: Describe how you prepare drugs for injection. Where do you get your water? What do you use to mix your drugs (spoon or bottle cap)? Does anyone else use the same thing (cooker) to mix their drugs?

Participant 11: We always use water from the tap to mix, if it is hard drug, we have to pound it on the spoon then we mix it with water. We get this water from the tap. After making a solution we put in our syringes that has a needle and our leader injects us. Not everyone knows how to inject drugs, you might end up injecting someone in the part which cannot be injected. However even on the stomach sometimes can be injected. (29-year-old man)

Some PWID sterilized their water for injecting by boiling it themselves, getting it preboiled from their dealers, or purchasing sterile water from pharmacies. However, sterile water bought in pharmacies was reported to be expensive.

Interviewer: How do you prepare the drugs for injection? Where do you get the water?

Participant 1: Where we get water from where we buy the drugs. They boil the water and put it in a tin. If you are nice to them they give you and we buy drip water (saline water) that is used for drips and not mixed with any other drugs. So, when we buy, and they measure and mix for us—we inject. That's why these drugs are expensive. If you can't afford the saline water, you use the local one. (20-year-old woman)

In the absence of any water source, some users reported using blood to prepare their drugs.

Interviewer: First, tell me how do you make the drug for injecting?

Participant 7: You get the drug, put it on a spoon, then mix it. Then, first get some blood from the veins, so when you don't have water, you mix using blood. (25-year-old man)

## Sex-related risks

Male participants reported getting money to buy drugs in a variety of ways including through work, (mainly in informal transportation (boda boda or motorcycle taxis), construction or street vending), through selling drugs or stealing. A sizeable number of women reported sex work or sex exchanges, even those who did not inject like the woman below.

Interviewer: Have you ever exchanged money for sex?

Participant 9: That's my job.

Interviewer: How about sex for drugs?

Participant 9: Sometimes, when I fail to get money on a bad day, I exchange sex for drugs. . ... At first, I did not like it but am now getting used. It's what I have to do to survive.

Interviewer: Why do you say you didn't like it the first time?

Participant 9: I thought it was bad; from the way people stigmatize us and the way they talk about us. But conditions pushed me to do it because I don't have any other job I can do.

Interviewer: Any other bad experience?

Participant 9: The body is now used and it's hard to stay without consuming the drug. Am sometimes forced to do bad things to get the drug. (31-year-old female)

Condom use in these encounters was rare. The women knew condoms would protect them from getting HIV but often reported forgetting to use one, especially if they were already high. Condom use also depended on whether the client was willing to pay more for condomless sex and the woman's desperation for drugs.

Men also reported exchanging drugs for sex if women did not have money to buy drugs as the man below reports. However, even though men also engaged in sex exchanges, their participation in this was much more sporadic when an opportunity presented itself as described by the man below. In contrast, women who considered sex work their jobs, often had several sexual partners in a day.

Interviewer: Ok, have you ever sold for someone marijuana or other drugs?

Participant 4: Yes.

Interviewer: How much were you paid?

Participant keeps quiet.

Interviewer: How much?

Participant 4: There is a woman I bought it for and she refused to pay me and decided to play sex with her and closed the chapter.

Interviewer: Meaning sometimes you exchange sex for drugs? The last time you bought for someone and exchanged sex, what do you recall that happened?

Participant 4: Yes,

Interviewer: You are only laughing, tell me what happened?

Participant 4: You see what happened, we played sex and without a condom, and I enjoyed. I think if everyone was playing sex without a condom, things would be sweet. (23-year-old male).

Condoms were often not used in these encounters as male clients preferred and would often pay more for condomless sex.

Interviewer: Have you ever exchanged sex for money, drug, or something else valuable?

Participant 25: Yes, I always exchange sex for money and the drug. Sex work has been my source of earning for many years and I have always been paid depending on the time spend with the client and the nature of sex to play [i.e. protected or not protected using a condom]. Meanwhile, the addiction to drugs means you cannot do without a drug. There are sometimes when I do not have required money to possess a drug or when I am with my friends to give me a shot. In this condition, I have asked for a drug from known drug

dealers and paid in kind. Most times in a bar, when a guy buys me alcohol they expect sex in return. (27-year-old female)

## Harm reduction services

As mentioned, the UHRN piloted a syringe exchange program in the year we conducted this study. Approximately three participants reported receiving syringes from an active drug user who gave syringes to people he knew. This included one woman and two men.

Interviewer: You told me about the injectables, heroin and cocaine; could you tell me where you obtain the syringes you use to inject drugs?

Participant 1: Am going to say but don't take him away. For syringes, we have [UHRN peer recruiter] and he is always supplying us syringes so we don't get sick. These days he is not bringing the syringes and I don't know why. That's why these days I don't inject and am missing. All I beg from the government is to help us because not all those who smoke, inject and sniff drugs are dangerous. We do it because of problems like having no money, standards of living, stress; joblessness leads us to that other than taking us to prisons. . . . . . . It's bad to take drugs but we are addicted. The government should help the addicts such that the syringes are supplied. We need more syringes and we are dying seriously. (20-year-old woman)

Serving as a peer health worker can have beneficial effects for the people who use drugs who often reduce their own risks or drug use to serve as a positive model for their peers. A UHRN peer syringe distributer described these benefits when describing his work.

Participant 7: I just mobilize drug users and when I do it, am given some money. We also organize blood testing and counselling programs for drug users. That's how I earn some money.

Interviewer: Which organizations are those?

Participant 7: Harm Reduction. . .. When I wake up in the morning, I take a shower and I know I have a program with so and so and I start making phone calls. That's how my day starts. So, whoever gives me a program then I have to fulfil it. I don't think about drugs all the time.

Interviewer: So, like in that schedule, when do you take the drugs?

Participant 7: . . .[W]hen I reach my place.

Interviewer: Why don't you take it anywhere apart from your place?

Participant 7: I am in the process of quitting drug use and I don't want other people to see me. If they see me on the street taking drugs, they will not respect me. . .. Like they think all the time you are high. Even when you are talking sense, they think you are high on drugs. (25-year-old man)

This participant obtained syringes from the UHRN to pass for free to his drug using peers. As mentioned, he took this role very seriously and avoided using drugs in front of his peers for fear they would not listen to his harm reduction advice, and because he was trying to cut down on his drug use. The fact that only three participants reported that they received syringes from an outreach worker suggests the need to expand the program. Unfortunately, the pilot lasted only a year and funding for it had just ended by the time the interviews for this project were conducted.

## HIV testing and condom distribution

While most participants reported having been tested for HIV, the frequency of testing was very low. Participants rarely sought out testing on their own; they were most commonly tested after being approached by outreach workers like UHRN trained users who took their peers to get HIV tests at UHRN centers.

Interviewer: Have you ever taken an HIV test? Where was it?

Participant 1: I went to Namwongo. It was group of us and we went with (UHRN trained peer educator who supplied syringes). Seriously am okay. This is where they pricked, and I was okay. With me I was okay, but others are sick. . . and others okay (negative). This all because of sharing the needles (20-year-old woman).

Other participants reported getting tested for HIV from organizations other than UHRN. These were not necessarily focused on PWID, but provided condoms, rapid testing and even information about anti-retroviral treatment (ART) and post-exposure prophylaxis for those who felt they may have been exposed to HIV.

Despite regular outreach to promote HIV testing and educate people who use drugs and the general population to test regularly for HIV, many refused to get tested. Some had heard that HIV is no longer a problem because everyone already has it and there is effective medication.

Interviewer: Have you heard your friends or other drug users discuss HIV? What have they said?

Participant 21: Yes, that HIV is nowadays a disease for everyone, it does not eliminate. They do not fear HIV since it does not kill customers. (33-year-old woman)

Conversely, others chose not to take the test because they are afraid of learning they are HIV positive.

Interviewers: Have you ever taken an HIV test?

Participant 9: It has been a long time. . .. I even fear because now I meet different men, so I don't know whether they have infected me or not. . .. The truth is that now I can't go for an HIV test. I will first wait and fall sick, then I go for a test. (31-year-old woman)

The woman reported that she "couldn't" go because she is too afraid and will only go if she is confronted by symptoms. Some reported that they knew people who would commit suicide if they found out they had HIV or try to infect others, indicating a high level of stigma regarding diagnosis and a lack of knowledge that one can live a long and healthy life with treatment.

Participant 14: Some. . . have this cheap thinking and reasoning that if he wakes up and is tested HIV positive, he commits suicide and another one was saying that in case he is tested and found to be HIV positive then he would infect like ten people before dying. (28-year-old man)

To date, PWID have not received HCV testing as part of the services they receive. As a result, it is not known how many are infected with HCV.

Participants reported that they had been offered condoms and educated about their use by outreach workers. However, condom use was still low due to participants forgetting, being high, or not liking condoms, even among those who had tested positive.

## Overdose prevention

Participants reported high rates of overdose from heroin use. As one participant said:

Participant 2: Actually, everyone who uses that drug of injection, he or she has overdosed because sometimes when you get money, you understand, you can go and shop as much as you want. After that you get too high and overdose. (26-year-old man)

Statistics regarding the number of fatal overdoses are not readily available.
While Naloxone is on Uganda's list of essential drugs, it is not yet available or accessible in the country. Thus, no participants users reported knowing what to do in cases of opioid overdose, although they sometimes reported using urine for alcohol poisoning.

Participant 31: We have also been also had cases of clients who are affected by overdose. And managing someone who has got overdose with especially hard drugs like brown sugar [heroin] is hard for us. With alcohol we have always managed overdose using human urine. (33-year-old woman)

Urine is not an effective treatment for opioid or alcohol overdoses. As the participant says, participants are left without effective means of reversing opioid overdoses.

## Quitting or reducing drug use

Very few participants reported trying to reduce or stop their drug use or of knowing anyone who had successfully quit using drugs. Many believed that quitting drugs was "impossible" because their bodies now needed the drug to function.

Interviewer: Ok, have you ever thought about stopping or reducing the use of drugs?

Participant 9: I can't stop.

Interviewer: Why?

Participant 9: It's because the body is now used. Even if it's after an hour of not taking the drug, I feel badly off. I have to look for money and take the drug. (31-year-old woman)

Participants who used drugs interpreted symptoms of withdrawal or craving as the body not being able to function without the drug. In part this may because people who use drugs have not been given information about drug treatment with the assistance of MAT or through common cognitive behavioral therapy approaches such as learning to cope with cravings and withdrawal. In fact, most participants claimed not to know anywhere where substance abuse treatment was offered or anyone who was successful in quitting.

Interviewer: Is there any person you know that stopped using drugs?

Participant 5: The one I know died. (30-year-old man)

Some participants mentioned prison as being a place where some temporarily abstained from drugs but claimed that most started their substance use as soon as they were released. Others mentioned the Butabika psychiatric hospital as the only place they knew of that people who use drugs were taken to recover from substance use. Without exception, participants reported that people taken to Butabika were mistreated and left in worse condition than when they entered.

Participant 1: You know, I fear being taken to rehabilitation. . .. One day my dad took me to Butabika, but life was hard. They injected me, and I would sleep the whole day. It's bad. When I came back, I said am not going back to my dad to take me to Butabika. In Butabika

we eat posho [food made from sorghum or millet flour with the consistency of dough]. Some patients have saliva coming out of their mouths. You can get mad [crazy] from there. We need a better place (20-year-old woman)

## Discussion

Results from our study demonstrate several risky injection practices in Uganda that could increase HIV incidence if actions are not taken. These are very similar to those found in other parts of the world, and include sharing syringes, sharing mixing containers, reusing syringes multiple times, using dirty water or blood to mix drugs and engaging in sex work [20–22]. While all participants reported knowing about the risks of contracting HIV by sharing syringes and other injection paraphernalia such as cookers and water, and syringes are available for sale without a prescription in pharmacies, the price of syringes and PWIDs' poverty made using a new syringe for each injection economically infeasible. According to a study conducted by Alliance Uganda and the UHRN [26], almost 40% of 125 PWID surveyed reported using syringes from two to four times, while 17.7% reported using them 10 or more times. Similarly, many PWID often had to pool resources to buy drugs, increasing the risk of sharing mixing water. These risks are common to all PWID in places without significant harm reduction efforts, such as syringe exchange programs, in locations were PWID can access them [27–30].

However, results from this study also reveal risks that may be unique to low- and middle-income countries, or to Uganda. First, the use of tap water to dilute drugs may increase the risk of abscesses, endocarditis, Hepatitis A and B and other infections even more than in developing countries because of the high levels of pathogenic viruses and bacteria in drinking water [31]. According to Water.org, 61% of Ugandans do not have access to safe drinking water as a result of rapid urbanization and poor sanitation infrastructure [32]. Large quantities of pathogenic bacteria have been found in springs that are the main source of drinking water for Kampala residents. Further, most PWID in Uganda live in informal settlements, or slum areas, many of which may have poorly designed pit latrines or lack sanitation facilities at all. Second, many participants reported buying premeasured drugs in syringes from drug dealers and then returning the used syringes to these dealers. It is not known whether these syringes are new or have been sterilized, but the willingness of PWIDs to buy drugs and inject with dealers' syringes is also a sign of the scarcity of clean needles in Kampala.

The UHRN pilot syringe exchange program was highly popular among users who received syringes in this way. A report of the SEP by UHRN reported that from July to September 2018, 2244 clean syringes were distributed to 120 PWID. Peers distributed syringes and kits containing cotton swabs, tourniquets, sterilized water, disposal bins and spoons. This program should be expanded to provide syringes and other safer injection materials to more PWID [18]. Peer-led interventions such as these have been shown to be successful in countries around the world where drug using outreach workers are able to access other people who use drugs that traditional outreach workers may not be able to [33]. These programs build on the altruism that people who use drugs have for other people who use drugs. Many peer-led interventions have shown that those who are trained to conduct outreach among their peers reduce their own risk behaviors, including drug use, to reduce cognitive dissonance between their own actions and the advice they give to others [33–36], much like the peer educator who tried to reduce his drug use in order to be credible and effective. Continued drug use among peer educators can also lead their drug using peers to assume that they are only doing outreach for the money which allows them to discount their messages [37]. The government of Uganda and UHRN have committed to expanding the program to drop-in centers (DICs) that will be in different parts of the country.

Naloxone should be provided as part of the kit of materials offered to PWID by their peers. Police, ambulance drivers and emergency department physicians should be trained in its use. Policy changes that will reduce the fear that people who use drugs may be arrested for trying to seek medical help for a peer who has overdosed are needed. Second, medication assisted therapy (MAT) should be made available to PWID and other opioid users who wish to stop. Buprenorphine or methadone could be offered in UHRN DICs since many people who use drugs around the world have expressed their preference for MAT in harm reduction facilities rather than medical clinics, due in part to the more tolerant and welcoming environments found in these locations [38]. In addition, the UHRN has reported that many people who use drugs avoid medical clinics because they fear being reported to police or otherwise face discrimination from staff [39]. While MAT is stigmatized among many users in the United States and other countries as people argue that it is replacing one drug with another [40], it may be more acceptable to people who use drugs in Uganda who reported needing medicine to help them abstain from the drugs that their bodies still need. However, psychosocial treatment is also needed as recognized by the participants in this study.

Results from this study suggest a near absence of any type of drug treatment, including abstinence only or 12-step programs. Most participants reported not knowing anyone who had successfully quit using drugs and did not know of any drug treatment facilities. A few had heard of people going to Butabika mental hospital for drug treatment and were afraid to go there as well. A study conducted by the Mental Disability Advocacy Centre observed a failure to provide for basic needs, such as sufficient food, and degrading treatment, involuntary confinement, seclusion and chemical restraints and a failure to provide biomedical or psychosocial treatment [41]. The participants' description of her time fits well with the reports observations and helps explain participants' fear of being sent to Butabika.

Additional services are needed to develop a combination prevention intervention that can effectively halt the spread of HIV and reduce other risks among PWID. First, efforts to increase rates of HIV testing among PWID are needed to link those with HIV into care. Few PWID in this study reported getting an HIV test more than once every few years, and some avoided testing altogether. Given the high rates of risk, and the prevalence of HIV rates among PWID estimated, HIV testing is a necessary first step to get PWID into HIV treatment and virally suppressed to prevent transmission to others. Social network HIV testing, in which PWID or other key populations are given monetary incentives to receive an HIV test and incentives for referring other members of their networks for an HIV test, has been very effective in increasing HIV testing among people who use crack cocaine in El Salvador and could increase testing rates in Uganda (Authors pub). Testing for Hepatitis C is needed to assess its prevalence among PWID and advocate for medications to treat it.

## Limitations

The present study has several limitations including the small sample size. Given that not all participants injected drugs, we may not have reached saturation of themes regarding drug injection risks. This study should be considered exploratory and its value lies in revealing the HIV risks of people who use injection and non-injection drugs in a setting that has only recently seen illicit drug use. However, the similarity of risks reported by participants within the study and in comparison to research conducted with people who inject drugs in other settings adds confidence to our results. In addition, experiences of people who used a variety of non-injection drugs may not have been adequately captured. The study was conducted in a number or drug hotspots in Kampala Uganda and may not represent the experiences of drug users in other parts of the country. However, because more harm reduction services are

available in Kampala than elsewhere in the country, we felt that it was important to document gaps in these efforts in Kampala as they are likely to be greater in other parts of the country.

## Conclusions

Injection drug use is a growing problem in Uganda and PWID are at high risk for HIV and HCV infection through injection and sexual risk behaviors. Aside from a small pilot that used peers to distribute syringes, PWID reported having to buy syringes from pharmacies which caused many to share due to the expense of syringes. A multi-component harm reduction intervention including syringes, naloxone, condoms, and MAT is needed to prevent HIV/ HCV infection in Uganda.

## Supporting information

**S1 File.**
(DOCX)

## Author Contributions

**Conceptualization:** Julia Dickson-Gomez, Ethan Homedi.

**Data curation:** Katende Dan, Ronald Anguzu, Ethan Homedi.

**Formal analysis:** Julia Dickson-Gomez, Erica Christenson, Ronald Anguzu, Ethan Homedi.

**Investigation:** Wamala Twaibu, Katende Dan.

**Project administration:** Wamala Twaibu, Katende Dan, Nazarius Mbona Tumwesigye.

**Resources:** Nazarius Mbona Tumwesigye.

**Supervision:** Katende Dan, Nazarius Mbona Tumwesigye.

**Writing – original draft:** Julia Dickson-Gomez.

**Writing – review & editing:** Julia Dickson-Gomez, Wamala Twaibu, Erica Christenson, Ronald Anguzu, Ethan Homedi.

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
