## [Decision Letter · Decision Letter 0]

4 Dec 2019

PONE-D-19-30396

Injection and Sexual Risk among People who Inject Drugs in Kampala, Uganda: A Qualitative Analysis

PLOS ONE

Dear Dr. Dickson-Gomez,

Thank you for submitting your manuscript to PLOS ONE. After careful consideration, we feel that it has merit but does not fully meet PLOS ONE’s publication criteria as it currently stands. Therefore, we invite you to submit a revised version of the manuscript that addresses the points raised during the review process.

We would appreciate receiving your revised manuscript by Jan 18 2020 11:59PM. To enhance the reproducibility of your results, we recommend that if applicable you deposit your laboratory protocols in protocols.io, where a protocol can be assigned its own identifier (DOI) such that it can be cited independently in the future. For instructions see: http://journals.plos.org/plosone/s/submission-guidelines#loc-laboratory-protocols

We look forward to receiving your revised manuscript.

Kind regards,

Joel Msafiri Francis, MD, MS, PhD

Academic Editor

PLOS ONE

Journal Requirements:

3. Please provide additional details regarding participant consent. In the ethics statement in the Methods and online submission information, please ensure that you have specified how verbal consent was documented and witnessed.

Reviewers' comments:

Reviewer's Responses to Questions

**Comments to the Author**

1. Is the manuscript technically sound, and do the data support the conclusions?

Reviewer #1: No

Reviewer #2: Yes

2. Has the statistical analysis been performed appropriately and rigorously? 

Reviewer #1: N/A

Reviewer #2: N/A

3. Have the authors made all data underlying the findings in their manuscript fully available?

Reviewer #1: Yes

Reviewer #2: Yes

4. Is the manuscript presented in an intelligible fashion and written in standard English?

Reviewer #1: No

Reviewer #2: Yes

5. Review Comments to the Author

Reviewer #1: Dickson-Gomez et al. explore injection and sexual risk behaviors among people who inject drugs in Kampala, Uganda. While certainly a worthy area of study, given the paucity of data from low-and-middle income countries with emerging injection drug use epidemics, this manuscript will need to be re-worked in several ways to ensure validity and robustness of findings and conclusions.

Major comments:

1. While I recognize that this is a particularly difficult population to work with, I am not certain why recruitment could not have been more purposive in that only injection drug users were recruited. Instead, it is misleading to state that 30 participants were recruited, and then go on to describe only the experiences of the 17 among them who were actually injecting for some parts of the manuscript and more general experiences of the other users in other parts of the manuscript. Infact this demarcation is not at all clear. It is also unclear if saturation of themes were achieved with these 17 interviews (among injecting PWID), which the authors themselves admit is a limitation. This is a major methodological flaw in the study. Instead the authors should aspire to conduct a more robust qualitative study, recruiting enough injecting (male and female) PWID to explore these areas in detail until saturation of themes is achieved. Respondent driven sampling (which the authors used partially) can be a powerful way to recruit participants. The experiences of male and female PWID also need to be more clearly delineated since there is enough data to suggest that HIV disproportionately affects female PWID. Barriers to receive harm reduction services also differ significantly between male and female PWID. Instead what appears in this manuscript is the experiences of male and female PWID inter-mingled such that the sexual risk behaviors among female PWID (who are also sex workers) is explored in detail, and yet little is known about the sexual risk behaviors of male PWID. The other alternative is to focus exclusively on female PWID (as has been done in other studies in other settings) as a separate manuscript.

2. The organization of the manuscript can also be significantly improved. Some suggestions are as follows:

a) The qualitative methodology needs to be explained in greater detail. Representative questions are presented (and can perhaps be organized under broad headings in a table); however not all questions are sufficiently dealt with in the results. For example, the authors state that they explored transitions to drug use and why injecting did or did not occur but do not describe this in the results.

b) Parent and child nodes for coding can be presented in a table.

c) At the very beginning, in the introduction, a pre-amble can be provided on the following: i) size estimates of PWID in Uganda (at least what is known), ii) existence of syringe exchange programs and opioid agonist therapy services, iii) targeted intervention programs for PWID, iv) voluntary testing and counseling centers, iii) government ART centers. While this data is offered in the manuscript, it is interspersed in the results and discussion. Whereas describing this information at the beginning provides clarity and helps the reader recognize what is available. Subsequent themes on barriers to receive services (including paucity of services) or other factors can then be appreciated in this context. The authors note that a "package of interventions" is recommended for PWID -- indeed, this is also the WHO's recommendation for key populations (given the challenges in reaching and linking such hard to reach populations). Presenting what is available in Uganda, and noting that as is the case in many countries, service delivery to key populations often tends to be fragmented (even if services are available) will offer context as well.

d) The quotes in the results section are interspersed with paragraphs such as data on clean water in Uganda. Firstly, for each theme that is explored, representative quotes can either be provided in text, or additional quotes can be provided in table format (or in an appendix). Secondly, the addition of explanatory data is distracting, and should only be reserved for the discussion section if relevant. The results section should be solely limited to the exploration of themes and presentation of quotes.

Minor comments:

1. There are numerous grammatical and spelling errors throughout the manuscript, and shifting from past tense to present tense within the same paragraph.

2. The authors note that they recruited 30 participants but state that they conducted 31 interviews. This is perhaps a typographical error?

3. Examples of well written qualitative papers exploring similar issues among PWID in low and middle income countries can serve as a guide for how to write this manuscript (see below for some references). Overall it is an important area of study. With more robust recruitment and methodology, this submission can emerge as a much stronger and valuable manuscript in the literature:

Kermode M et al. Harm Reduct J. 2007 Dec 5;4:19. My first time: initiation into injecting drug use in Manipur and Nagaland, north-east India.

Kermode M et al. BMC Int Health Hum Rights. 2013 Jan 29;13:9. Falling through the cracks: a qualitative study of HIV risks among women who use drugs and alcohol in Northeast India.

Reviewer #2: Overall, this work helps fill an important gap in the prevention literature, which is severely lacking for this country. The work is presented cohesively. I do not have any major comments. Minor suggestions regarding syntax and grammar follow below.

Methods:

• Correct “people who used illicit drug use” in the first sentence

• Change “assess” to past tense

• Compound adjectives should be hyphenated. For example, “injection and drug related sexual risk” should be “injection- and drug-related sexual risk”.

• “HIV risky” should be “HIV risk”

Results

• The sentence “As a result, nearly all participants who injected drugs reported sharing injection equipment” is placed a bit prematurely in the paragraph—it only becomes clear why participants were reusing syringes in the following sentences. I would simply remove “As a result” to circumvent this issue. This also prevents repetition of this phrase within the paragraph.

• You may want to consider labelling quotes with “Interviewer” and “Participant ##” at the very least, to help the reader distinguish between speakers. Labelling by participant number also shows the reader that the quotes are not all derived from a small number of participants. There is some labelling throughout the section, but it is inconsistent.

• The paragraph that begins with “In this situation of scarcity” is a run-on sentence. Consider breaking this up into separate sentences to make it more digestible.

• The information about an external study conducted by the UHRN should not be presented in the results section. It can be linked to the results in the discussion section, as appropriate. Similarly, any discussion about the implications of the results, or framing context, should not be introduced in the results section.

• In the sentence which contains “reported returning syringes to users after injecting”, did the authors mean to say that syringes were returned to sellers after injecting?

Discussion/Conclusion

• There are some acronyms used in the conclusion which are not defined anywhere.

• Much of the conclusion seems to be discussion about implications of this research, rather than a conclusion proper. I would suggest restructuring this content so that the conclusion is more succinct, and the discussion includes the implications of this research and suggestions for future work.

6. PLOS authors have the option to publish the peer review history of their article (what does this mean?). If published, this will include your full peer review and any attached files.

Reviewer #1: No

Reviewer #2: Yes: Alberto Edeza

---

## [Author Response · Author response to Decision Letter 0]

2 Jan 2020

Thank you for your careful review of our manuscript “Injection and sexual risk among people who use or inject drugs in Kampala, Uganda: A qualitative analysis.” The reviewers thought there was merit in the paper, but suggested a number of revisions which we outline below. Unfortunately, I did not read that we should submit a copy with track changes and one without, so I have highlighted the parts that were changed in the revised manuscript “with tracked changes.” We also address editor comments regarding format, data accessibility and making supplementary material such as interview guides available.

“While I recognize that this is a particularly difficult population to work with, I am not certain why recruitment could not have been more purposive in that only injection drug users were recruited. Instead, it is misleading to state that 30 participants were recruited, and then go on to describe only the experiences of the 17 among them who were actually injecting for some parts of the manuscript and more general experiences of the other users in other parts of the manuscript. Infact this demarcation is not at all clear. It is also unclear if saturation of themes were achieved with these 17 interviews (among injecting PWID), which the authors themselves admit is a limitation. This is a major methodological flaw in the study. Instead the authors should aspire to conduct a more robust qualitative study, recruiting enough injecting (male and female) PWID to explore these areas in detail until saturation of themes is achieved. Respondent driven sampling (which the authors used partially) can be a powerful way to recruit participants. The experiences of male and female PWID also need to be more clearly delineated since there is enough data to suggest that HIV disproportionately affects female PWID. Barriers to receive harm reduction services also differ significantly between male and female PWID. Instead what appears in this manuscript is the experiences of male and female PWID inter-mingled such that the sexual risk behaviors among female PWID (who are also sex workers) is explored in detail, and yet little is known about the sexual risk behaviors of male PWID. The other alternative is to focus exclusively on female PWID (as has been done in other studies in other settings) as a separate manuscript.” We believe that it was important, given the scarcity of knowledge about illicit drug use in Uganda, to include both PWID and PWUD but did not inject. Both groups engaged in sex exchange and other sexual-risk behaviors as we now clarify. The additional risk faced by women PWID and PWUD are in the frequency in which they exchanged sex, as many considered sex work as their main means of support. We believe that we received saturation of themes for injectors and non-injectors, women and men.

Reviewer 1 suggested several improvements in organization, including the following. 

“The qualitative methodology needs to be explained in greater detail. Representative questions are presented (and can perhaps be organized under broad headings in a table); however not all questions are sufficiently dealt with in the results. For example, the authors state that they explored transitions to drug use and why injecting did or did not occur but do not describe this in the results.” We have taken out descriptions of questions which were not analyzed or presented in this paper. The entire interview guide is included in supplementary materials.

“Parent and child nodes for coding can be presented in a table.” We have included the coding tree in a Table in the manuscript.

“At the very beginning, in the introduction, a pre-amble can be provided on the following: i) size estimates of PWID in Uganda (at least what is known), ii) existence of syringe exchange programs and opioid agonist therapy services, iii) targeted intervention programs for PWID, iv) voluntary testing and counseling centers, iii) government ART centers. While this data is offered in the manuscript, it is interspersed in the results and discussion. Whereas describing this information at the beginning provides clarity and helps the reader recognize what is available. Subsequent themes on barriers to receive services (including paucity of services) or other factors can then be appreciated in this context. The authors note that a "package of interventions" is recommended for PWID -- indeed, this is also the WHO's recommendation for key populations (given the challenges in reaching and linking such hard to reach populations). Presenting what is available in Uganda, and noting that as is the case in many countries, service delivery to key populations often tends to be fragmented (even if services are available) will offer context as well.” The first paragraph of the paper included the size of the PWID population in Uganda, existence of syringe exchange programs and MAT, and targeted intervention programs for PWID. We have included information on government ART centers and HIV testing through UHRN. We have also described the developmental phase of these programs, and the lack of coordinated services in most LMICs.

“The quotes in the results section are interspersed with paragraphs such as data on clean water in Uganda. Firstly, for each theme that is explored, representative quotes can either be provided in text, or additional quotes can be provided in table format (or in an appendix). Secondly, the addition of explanatory data is distracting, and should only be reserved for the discussion section if relevant. The results section should be solely limited to the exploration of themes and presentation of quotes.” This was also a comment made by Reviewer 2. We have taken out all references to previously published research and implications of results to the Discussion section.

“There are numerous grammatical and spelling errors throughout the manuscript, and shifting from past tense to present tense within the same paragraph.” We have corrected these errors.

“The authors note that they recruited 30 participants but state that they conducted 31 interviews. This is perhaps a typographical error?” Indeed, this was a typographical error. We conducted 30 interviews and this is now consistent throughout.

“Examples of well written qualitative papers exploring similar issues among PWID in low and middle income countries can serve as a guide for how to write this manuscript (see below for some references). Overall it is an important area of study. With more robust recruitment and methodology, this submission can emerge as a much stronger and valuable manuscript in the literature:

Kermode M et al. Harm Reduct J. 2007 Dec 5;4:19. My first time: initiation into injecting drug use in Manipur and Nagaland, north-east India.

Kermode M et al. BMC Int Health Hum Rights. 2013 Jan 29;13:9. Falling through the cracks: a qualitative study of HIV risks among women who use drugs and alcohol in Northeast India.”

We thank the reviewer for this suggestion and carefully read the manuscripts.

From Reviewer 2:

“Methods:

• Correct “people who used illicit drug use” in the first sentence

• Change “assess” to past tense

• Compound adjectives should be hyphenated. For example, “injection and drug related sexual risk” should be “injection- and drug-related sexual risk”.

• “HIV risky” should be “HIV risk”

We have made all these changes.

Results

• The sentence “As a result, nearly all participants who injected drugs reported sharing injection equipment” is placed a bit prematurely in the paragraph—it only becomes clear why participants were reusing syringes in the following sentences. I would simply remove “As a result” to circumvent this issue. This also prevents repetition of this phrase within the paragraph.

We have followed this suggestion.

• You may want to consider labelling quotes with “Interviewer” and “Participant ##” at the very least, to help the reader distinguish between speakers. Labelling by participant number also shows the reader that the quotes are not all derived from a small number of participants. There is some labelling throughout the section, but it is inconsistent.

We have followed this suggestion and include Participant ID numbers to distinguish participants.

• The paragraph that begins with “In this situation of scarcity” is a run-on sentence. Consider breaking this up into separate sentences to make it more digestible.

We have broken up this sentence.

• The information about an external study conducted by the UHRN should not be presented in the results section. It can be linked to the results in the discussion section, as appropriate. Similarly, any discussion about the implications of the results, or framing context, should not be introduced in the results section

We have removed all instances of external studies referenced in the results section and moved them to the Discussion. This includes the information about water quality, the effectiveness of peer led interventions as well as the UHRN study mentioned.

• In the sentence which contains “reported returning syringes to users after injecting”, did the authors mean to say that syringes were returned to sellers after injecting?

Yes, we have corrected this mistake.

Discussion/Conclusion

• There are some acronyms used in the conclusion which are not defined anywhere.

• Much of the conclusion seems to be discussion about implications of this research, rather than a conclusion proper. I would suggest restructuring this content so that the conclusion is more succinct, and the discussion includes the implications of this research and suggestions for future work.

We have moved much of what was formerly in conclusions to the discussion section as the information in conclusions discussed implications of findings. We have also defined all acronyms. The conclusion now only contains a brief summary of findings.

We hope that these changes adequately address reviewers’ concerns.

Sincerley,

Julia Dickson-Gomez

Professor of Epidemiology

Institute for Health and Equity

Medical College of Wisconsin

---

## [Decision Letter · Decision Letter 1]

23 Jan 2020

PONE-D-19-30396R1

Injection and sexual risk among people who use or inject drugs in Kampala, Uganda:  A qualitative analysis

PLOS ONE

Dear Dr. Dickson-Gomez,

Thank you for submitting your manuscript to PLOS ONE. After careful consideration, we feel that it has merit but does not fully meet PLOS ONE’s publication criteria as it currently stands. Therefore, we invite you to submit a revised version of the manuscript that addresses the points raised during the review process.

We would appreciate receiving your revised manuscript by Mar 08 2020 11:59PM. To enhance the reproducibility of your results, we recommend that if applicable you deposit your laboratory protocols in protocols.io, where a protocol can be assigned its own identifier (DOI) such that it can be cited independently in the future. For instructions see: http://journals.plos.org/plosone/s/submission-guidelines#loc-laboratory-protocols

We look forward to receiving your revised manuscript.

Kind regards,

Joel Msafiri Francis, MD, MS, PhD

Academic Editor

PLOS ONE

Reviewers' comments:

Reviewer's Responses to Questions

**Comments to the Author**

1. If the authors have adequately addressed your comments raised in a previous round of review and you feel that this manuscript is now acceptable for publication, you may indicate that here to bypass the “Comments to the Author” section, enter your conflict of interest statement in the “Confidential to Editor” section, and submit your "Accept" recommendation.

Reviewer #1: (No Response)

Reviewer #2: All comments have been addressed

2. Is the manuscript technically sound, and do the data support the conclusions?

Reviewer #1: Partly

Reviewer #2: Yes

3. Has the statistical analysis been performed appropriately and rigorously? 

Reviewer #1: Yes

Reviewer #2: N/A

4. Have the authors made all data underlying the findings in their manuscript fully available?

Reviewer #1: No

Reviewer #2: No

5. Is the manuscript presented in an intelligible fashion and written in standard English?

Reviewer #1: Yes

Reviewer #2: Yes

6. Review Comments to the Author

Reviewer #1: Dr Dickinson-Gomez and colleagues have submitted a revised manuscript that explores injection and sexual risk behaviors, and access of harm reduction services among people who use drugs in Uganda. Overall, this revised version is an improvement from the initial submission.

However, I do think that the manuscript warrants at least another round of major revisions in order to make it both suitable for publication and valuable to readers.

MAJOR Comments:

1. In this revision, the authors have been more transparent about their rationale to recruit both people who inject drugs and people who use drugs; the study also clearly includes both genders which is certainly valuable. In their response to my concern that the small number of participants may not have facilitated achievement of saturation of themes, the author state the following: “We believe that we received saturation of themes for injectors and non-injectors, women and men”. I want to point out that this directly contradicts what they write in the limitations section of this manuscript: “The present study has several limitations including the small sample size characteristic of qualitative studies. Given that not all participants injected drugs, we may not have reached saturation of themes regarding drug injection risks”. Also, I should note that nowhere in their methodology have they stated that they actually continued recruitment till saturation of themes was achieved.

I would suggest the following changes:

a. Small sample sizes are not a characteristic of qualitative studies. There are qualitative studies that include several hundred participants. It is best not to suggest this. What is true however is that, in general most investigators would halt recruitment when they start hearing the same themes (i.e., achieve saturation).

b. So instead of saying “we may not have achieved saturation of themes for certain topics”, and then stating (at least in their response ) that they did in fact achieve saturation, it’s best to perhaps characterize this as an exploratory qualitative study that despite its small numbers offers insights into risk behaviors among people in a neglected key population in need of services. The title of the manuscript can also be reworded as such: “Injection and sexual risk among people who use or inject drugs in Kampala, Uganda: an exploratory qualitative study”.

2. While the results section has been reorganized, I think this section can further be improved. Again, explanations for certain observations is not needed in this section. Please see below for suggestions on how to improve this section.

3. Throughout the manuscript, the term “availability” of harm reduction services is used. I think what the authors are actually assessing is whether PWID accessed harm reduction services. As they state in the introduction, there are few if any harm reduction services that are available. What is also true is that few PWID are actually accessing even the available services.

4. Acronyms are used throughout the manuscript. However, they should be described in full the very first time they are used. Please ensure this is the case. Also, after an acronym is already used, there are places where full forms are still presented – for example: Syringe exchange programs.

MINOR Comments:

1. Abstract:

Background: Suggest modifying last statement as: “The objective of the research was to document injection and other drug related-risk behaviors among people who use drugs in Uganda to develop comprehensive HIV/HCV prevention intervention.

Methods: This section needs more details on qualitative methodology. Again, the term “availability” is used to describe both injecting equipment and harm reduction services. I believe the authors meant to say PWUD/PWID access of harm reduction and drug treatment services.

Results: Should include the demographic characteristics at least briefly – how many were recruited etc. UHRN – acronym is used. Please provide full form following which acronym can be used in the rest of the manuscript.

Conclusions: Comprehensive (?drug use) prevention and treatment services. “Linkage and retention to HIV care” should be reworded for grammar: “ Linkage to and retention in HIV care”.

2. Introduction:

transshipment points: Trans-shipment (with a hyphen)

PEPFAR COP2018 – While many are familiar with PEPFAR, what does COP stand for?

“Most researchers argue that a combination of approaches are needed including: provision of clean syringes through syringe exchange programs (SEPs); frequent HIV testing and linkage to HIV care; antiretroviral therapy initiation after infection to reduce risk of onward transmission through viral suppression; medication assisted therapy; psychosocial support and pre- and post-exposure prophylaxis” : A colon is not needed after including and every point can be separated by commas.

“harm reduction centers to the rest of the country and include MAT, syringe exchange and PrEP”: sentence should be rephrased as “ harm reduction centers to the rest of the country that include services such as MAT, syringe exchange and PrEP.

“The current study presents results from qualitative interviews with 30 people in Kampala, Uganda who use injection and non-injection drugs to determine their behavioral risks for HIV and the availability of resources to reduce the harms associated with injection such as clean syringes, non-pharmaceutical drug treatment, condoms and HIV testing”. I am not sure what non-pharmaceutical drug treatment actually means? Buprenorphine for example is a pharmaceutical drug and is used in MAT programs.

“We include both people who inject drugs (PWID, n=17) and people who use drugs (PWUD) but do not inject (n=14), because while injection with contaminated syringes is one of the most efficient ways to become infected with HIV, PWID and PWUD can be at risk for becoming infected through highrisk sexual behaviors. In addition, PWID and PWUD often are parts of the same drug using networks”. This should be moved to methods.

3. Eligibility criteria and interview sampling characteristics

“We purposively oversampled to include more women than in the general drug using population, and injection and non-injection users and from each community”. This sentence is not clear and am not sure what this is supposed to mean. With overall small sample, I am not sure this is accurate either.

4. In-depth interview procedures and content:

“any experiences with harm reduction services such as clean needles and HIV testing”: I think the authors meant to say, “harm reduction services such as SEP and HIV testing”.

5. Qualitative analysis: The coding tree while helpful has some categories that clearly are based on the interview guide, and other terms that appear more random.

6. Results:

This section should just start with the descriptive details of patient characteristics. Other sections go on to elaborate on the various risk behaviors – so a summary does not need to be provided here.

Headings need to be broader and consistent with what you sought out to explore. For example, Syringe sharing, use of unclean water/ mixing practices will fall under a broader category of Injection related risk behaviors.

This entire section is written in present tense and is not consistent with the tense used in the rest of the manuscript

There are some typographical errors: “Some PWID sterilized their water for injecting by boiling it themselves, getting in pre-boiled from their dealers, or purchasing sterile water from pharmacies”. I think the authors meant to say “getting it pre-boiled”.

Again, please avoid providing explanations or consequences in the results section. For example: “If drugs, syringes, or spoons are shared after mixing drugs with blood, the risk for contracting HIV/HCV is high”.

Harm reduction services: There is suddenly a mention of an individual by name of “Ronald”—I am presuming that this was actually participant 7 who is identified as a peer syringe distributer and then is mentioned as being Ronald. It may suffice to just state that those who were peer educators experienced benefits without having to name Ronald.

Overdose prevention: “Workers at the UHRN and participants confirmed high rates of overdose from heroin use”. Theoretically, workers at UHRN were not part of this study, unless they also participated in interviews. It may be better to present overdose data collected by UHRN in the discussion if available and stick to just participant descriptions in this section

Quitting or reducing drug use: “In part this may because people who use drugs have not been given information about drug treatment, either through MAT, or by coping with withdrawal and cravings in the case of cocaine addiction”. This sentence is grammatically incorrect.

7. Discussion:

“While harm reduction efforts have mainly focused on PWID in the developing world, injecting with contaminated water may increase the risk of abscesses, sepsis, endocarditis or other infections associated with injection” – This is a repetition (already mentioned earlier in the paragraph).

Reviewer #2: Dear Author,

This manuscript presents a significant contribution to the literature around injection-related risk behaviors for HIV and other blood borne pathogens among PWID and PWUD in Uganda. As a formative study, this qualitative work paints a broad picture of the context of drug use among some PWID and PWUD in Uganda, and clearly delineates avenues that should be pursued in further work. The authors addressed all of my comments. I recommend that the manuscript be accepted.

A small note--during proofing, in the first sentence of the conclusion, change 'in' to 'is'.

7. PLOS authors have the option to publish the peer review history of their article (what does this mean?). If published, this will include your full peer review and any attached files.

Reviewer #1: No

Reviewer #2: Yes: Alberto Edeza

---

## [Author Response · Author response to Decision Letter 1]

18 Feb 2020

Dear Dr. Msafirir Francis and Reviewers,

Thank you for your careful review of the paper which was found to be improved but still in need of some clarification, particularly with regards to whether we achieved data saturation. We agree with Reviewer 1’s comments and do not claim to have reached data saturation and changed the title and limitations accordingly. We have also included access as well as availability of harm reduction materials where appropriate and corrected typos and grammatical errors.

In this revision, the authors have been more transparent about their rationale to recruit both people who inject drugs and people who use drugs; the study also clearly includes both genders which is certainly valuable. In their response to my concern that the small number of participants may not have facilitated achievement of saturation of themes, the author state the following: “We believe that we received saturation of themes for injectors and non-injectors, women and men”. I want to point out that this directly contradicts what they write in the limitations section of this manuscript: “The present study has several limitations including the small sample size characteristic of qualitative studies. Given that not all participants injected drugs, we may not have reached saturation of themes regarding drug injection risks”. Also, I should note that nowhere in their methodology have they stated that they actually continued recruitment till saturation of themes was achieved.

I would suggest the following changes:

a. Small sample sizes are not a characteristic of qualitative studies. There are qualitative studies that include several hundred participants. It is best not to suggest this. What is true however is that, in general most investigators would halt recruitment when they start hearing the same themes (i.e., achieve saturation).

b. So instead of saying “we may not have achieved saturation of themes for certain topics”, and then stating (at least in their response ) that they did in fact achieve saturation, it’s best to perhaps characterize this as an exploratory qualitative study that despite its small numbers offers insights into risk behaviors among people in a neglected key population in need of services. The title of the manuscript can also be reworded as such: “Injection and sexual risk among people who use or inject drugs in Kampala, Uganda: an exploratory qualitative study”.

We have made the suggested change in wording in the limitations section and the title of the paper. 

While the results section has been reorganized, I think this section can further be improved. Again, explanations for certain observations is not needed in this section. Please see below for suggestions on how to improve this section.. Throughout the manuscript, the term “availability” of harm reduction services is used. I think what the authors are actually assessing is whether PWID accessed harm reduction services. As they state in the introduction, there are few if any harm reduction services that are available. What is also true is that few PWID are actually accessing even the available services.

We have included accessibility along with availability when appropriate. For example, HCV testing and Naloxone is currently not available. Clean syringes, HIV testing and condoms are available but not always accessible or accessed. We have tried to clarify these differences throughout.

Acronyms are used throughout the manuscript. However, they should be described in full the very first time they are used. Please ensure this is the case. Also, after an acronym is already used, there are places where full forms are still presented – for example: Syringe exchange programs.

We have spelled out all acronyms the first time they appear.

MINOR Comments:

Abstract:

Background: Suggest modifying last statement as: “The objective of the research was to document injection and other drug related-risk behaviors among people who use drugs in Uganda to develop comprehensive HIV/HCV prevention intervention.

We have edited the sentence as suggested.

Methods: This section needs more details on qualitative methodology. Again, the term “availability” is used to describe both injecting equipment and harm reduction services. I believe the authors meant to say PWUD/PWID access of harm reduction and drug treatment services.

We mean both availability and accessibility and have included both terms. As stated before, some services are currently not available like MAT, Naloxone, or HCV testing, others are available but so difficult to access as to make them virtually unavailable (e.g. the availability of clean syringes in pharmacies but their unaffordability, and the unavailability of SEPs except for the pilot study.

Results: Should include the demographic characteristics at least briefly – how many were recruited etc. UHRN – acronym is used. Please provide full form following which acronym can be used in the rest of the manuscript.

Demographics of the sample are included in a table. Acronyms are spelled out when first used. 

Conclusions: Comprehensive (?drug use) prevention and treatment services. “Linkage and retention to HIV care” should be reworded for grammar: “ Linkage to and retention in HIV care”.

We have corrected this error

Introduction:

transshipment points: Trans-shipment (with a hyphen)

We have corrected this error

PEPFAR COP2018 – While many are familiar with PEPFAR, what does COP stand for?

COP stands for Country Operational Plan. We omitted this since it did not seem necessary for comprehension.

“Most researchers argue that a combination of approaches are needed including: provision of clean syringes through syringe exchange programs (SEPs); frequent HIV testing and linkage to HIV care; antiretroviral therapy initiation after infection to reduce risk of onward transmission through viral suppression; medication assisted therapy; psychosocial support and pre- and post-exposure prophylaxis” : A colon is not needed after including and every point can be separated by commas.

We have made this change.

“harm reduction centers to the rest of the country and include MAT, syringe exchange and PrEP”: sentence should be rephrased as “ harm reduction centers to the rest of the country that include services such as MAT, syringe exchange and PrEP.

We have made this change.

“The current study presents results from qualitative interviews with 30 people in Kampala, Uganda who use injection and non-injection drugs to determine their behavioral risks for HIV and the availability of resources to reduce the harms associated with injection such as clean syringes, non-pharmaceutical drug treatment, condoms and HIV testing”. I am not sure what non-pharmaceutical drug treatment actually means? Buprenorphine for example is a pharmaceutical drug and is used in MAT programs.

Non-pharmaceutical treatment includes Cognitive behavioral therapy, or even self-help groups and 12-step programs. We included these because MAT was not yet available in Uganda.

“We include both people who inject drugs (PWID, n=17) and people who use drugs (PWUD) but do not inject (n=14), because while injection with contaminated syringes is one of the most efficient ways to become infected with HIV, PWID and PWUD can be at risk for becoming infected through highrisk sexual behaviors. In addition, PWID and PWUD often are parts of the same drug using networks”. This should be moved to methods.

We moved this to the Methods Section.

Eligibility criteria and interview sampling characteristics

“We purposively oversampled to include more women than in the general drug using population, and injection and non-injection users and from each community”. This sentence is not clear and am not sure what this is supposed to mean. With overall small sample, I am not sure this is accurate either.

We meant that we recruited both injection and non-injection drug users and attempted to oversample women to get their perspectives. We did not attempt to sample from each community and we have corrected this misunderstanding.

In-depth interview procedures and content:

“any experiences with harm reduction services such as clean needles and HIV testing”: I think the authors meant to say, “harm reduction services such as SEP and HIV testing”.

You are correct and we have made this change.

5. Qualitative analysis: The coding tree while helpful has some categories that clearly are based on the interview guide, and other terms that appear more random.

We now highlight that the coding tree came directly from research questions as well as unexpected themes in the data such as Butabika, the mental hospital where some had received drug treatment, and which was widely feared by drug users. Similarly, we added drug confiscation as this was reported as drug users reported that police frequently confiscated drugs.

6. Results:

This section should just start with the descriptive details of patient characteristics. Other sections go on to elaborate on the various risk behaviors – so a summary does not need to be provided here.

We eliminated a description of the risk behaviors and only include the participant characteristics.

Headings need to be broader and consistent with what you sought out to explore. For example, Syringe sharing, use of unclean water/ mixing practices will fall under a broader category of Injection related risk behaviors.

We have put Syringe Sharing and Use of Unclean Water/Mixing Practices under Injection Risk. We relabeled Sex Exchange and Sex-Related Risk behaviors.

This entire section is written in present tense and is not consistent with the tense used in the rest of the manuscript

Most of the section was written in the past tense with some general statements such as “Naloxone is not currently available” in the present tense. We have corrected the few instances in which this was not the case.

There are some typographical errors: “Some PWID sterilized their water for injecting by boiling it themselves, getting in pre-boiled from their dealers, or purchasing sterile water from pharmacies”. I think the authors meant to say “getting it pre-boiled”.

We believe we have caught and corrected the typographical errors.

Again, please avoid providing explanations or consequences in the results section. For example: “If drugs, syringes, or spoons are shared after mixing drugs with blood, the risk for contracting HIV/HCV is high”.

We have eliminated this.

Harm reduction services: There is suddenly a mention of an individual by name of “Ronald”—I am presuming that this was actually participant 7 who is identified as a peer syringe distributer and then is mentioned as being Ronald. It may suffice to just state that those who were peer educators experienced benefits without having to name Ronald.

Ronald, as mentioned in the original manuscript, was a pseudonym. However, we have changed the quotes to say [UHRN peer educator]. I do not know if Participant 7 and “Ronald” are the same person as we did not collect personal information like names.

Overdose prevention: “Workers at the UHRN and participants confirmed high rates of overdose from heroin use”. Theoretically, workers at UHRN were not part of this study, unless they also participated in interviews. It may be better to present overdose data collected by UHRN in the discussion if available and stick to just participant descriptions in this section

We have made this change.

Quitting or reducing drug use: “In part this may because people who use drugs have not been given information about drug treatment, either through MAT, or by coping with withdrawal and cravings in the case of cocaine addiction”. This sentence is grammatically incorrect.

We have corrected this sentence.

7. Discussion:

“While harm reduction efforts have mainly focused on PWID in the developing world, injecting with contaminated water may increase the risk of abscesses, sepsis, endocarditis or other infections associated with injection” – This is a repetition (already mentioned earlier in the paragraph).

We have eliminated this redundancy.

Reviewer #2: Dear Author,

This manuscript presents a significant contribution to the literature around injection-related risk behaviors for HIV and other blood borne pathogens among PWID and PWUD in Uganda. As a formative study, this qualitative work paints a broad picture of the context of drug use among some PWID and PWUD in Uganda, and clearly delineates avenues that should be pursued in further work. The authors addressed all of my comments. I recommend that the manuscript be accepted.

A small note--during proofing, in the first sentence of the conclusion, change 'in' to 'is'.

 We have corrected this typo.

We hope that the revised manuscript has addressed all remaining concerns. 

Sincerely,

Julia Dickson-Gomez

---

## [Decision Letter · Decision Letter 2]

31 Mar 2020

PONE-D-19-30396R2

Injection and sexual risk among people who use or inject drugs in Kampala, Uganda:   An exploratory qualitative study

PLOS ONE

Dear Dr. Dickson-Gomez,

Thank you for submitting your manuscript to PLOS ONE. After careful consideration, we feel that it has merit but does not fully meet PLOS ONE’s publication criteria as it currently stands. Therefore, we invite you to submit a revised version of the manuscript that addresses the points raised during the review process.

Thank you very much for working on the suggested revisions. Please kindly address reviewer -1 additional suggestions.

We would appreciate receiving your revised manuscript by May 15 2020 11:59PM. To enhance the reproducibility of your results, we recommend that if applicable you deposit your laboratory protocols in protocols.io, where a protocol can be assigned its own identifier (DOI) such that it can be cited independently in the future. For instructions see: http://journals.plos.org/plosone/s/submission-guidelines#loc-laboratory-protocols

We look forward to receiving your revised manuscript.

Kind regards,

Joel Msafiri Francis, MD, MS, PhD

Academic Editor

PLOS ONE

Additional Editor Comments (if provided):

Thank you very much for working on the suggested revisions. Please kindly address reviewer -1 additional suggestions.

Reviewers' comments:

Reviewer's Responses to Questions

**Comments to the Author**

1. If the authors have adequately addressed your comments raised in a previous round of review and you feel that this manuscript is now acceptable for publication, you may indicate that here to bypass the “Comments to the Author” section, enter your conflict of interest statement in the “Confidential to Editor” section, and submit your "Accept" recommendation.

Reviewer #1: All comments have been addressed

Reviewer #2: All comments have been addressed

2. Is the manuscript technically sound, and do the data support the conclusions?

Reviewer #1: Yes

Reviewer #2: Yes

3. Has the statistical analysis been performed appropriately and rigorously? 

Reviewer #1: Yes

Reviewer #2: N/A

4. Have the authors made all data underlying the findings in their manuscript fully available?

Reviewer #1: Yes

Reviewer #2: No

5. Is the manuscript presented in an intelligible fashion and written in standard English?

Reviewer #1: Yes

Reviewer #2: Yes

6. Review Comments to the Author

Reviewer #1: Please see attached document.

I only have minor comments for the authors.

Abstract:

“ Availability and accessibility of syringes” – amended to “ Availability and accessibility of clean needles and syringes”

Introduction:

“While Uganda has been the site of abundant prevention and research efforts” – include the term “ HIV prevention and research efforts”.

“it is one of only two countries in Sub-Saharan Africa (SSA) in which incidence is increasing” – include the term “ HIV incidence”.

“In Kenya, of 101 current injectors, 52.5% were found to be HIV positive, attributed mainly to sharing injection equipment compared to 13.5% of heroin users who had not injected” – include reference

“In 2018, the Kampala Place Study estimated over 18,000 people who inject drugs (PWID) in Uganda” – include reference

“12-step prgrams as MAT was not available at the time of study” – typographical error in programs. “As MAT was not available at the time of study “ should be in parentheses

Methods:

“Some codes, such as Butabika and drug confiscation by police authorities emerged from the interviews and were added after careful reading of transcripts”. This can be re-written to just say that new themes that emerged were included in the final coding scheme.

Results:

Injection risk behaviors:

“In this situation of scarcity, even people who know the risks of sharing and take precautions to protect themselves reported that they would share their used syringes with other users” – modify tense: In this situation of scarcity, even people who knew the risks of sharing and took precautions to protect themselves reported that they would share their used syringes with other users.

“Syringes were also be shared when PWID bought premeasured syringes from drug dealers” – grammatical error. Should be modified as “Syringes were also shared when PWID bought premeasured syringes from drug dealers”.

“Unfortunately, the pilot lasted only a year and funding for it had just ended by the time the interviews for this project were conducted, and many participants reported that they had not seen Ronald or received syringes from him for some time”. Ronald is suddenly introduced here. I think this can just be limited to saying, “Unfortunately, the pilot lasted only a year and funding for it had just ended by the time the interviews for this project were conducted”.

“Some have heard that HIV is no longer a problem because everyone already has it and there is effective medication”. Change to past tense for consistency with rest of paragraph: Some had heard that HIV was no longer a problem because everyone already had it and there was effective medication.

“The woman reported that she “can’t” go because she is too afraid and will only go if she is confronted by symptoms”. Again, change to past tense

Reviewer #2: (No Response)

7. PLOS authors have the option to publish the peer review history of their article (what does this mean?). If published, this will include your full peer review and any attached files.

Reviewer #1: No

Reviewer #2: Yes: Alberto Edeza

---

## [Author Response · Author response to Decision Letter 2]

1 Apr 2020

Thank you for the careful reading of the manuscript. We have made all the edits suggested by the reviewer outlined below. 

I only have minor comments for the authors. Would recommend that the journal proof-read for grammar and typographical errors. 

Abstract: 

“ Availability and accessibility of syringes” – amended to “ Availability and accessibility of clean needles and syringes” We have made this change.

Introduction:

“While Uganda has been the site of abundant prevention and research efforts” – include the term “ HIV prevention and research efforts”. We have made this change.

“it is one of only two countries in Sub-Saharan Africa (SSA) in which incidence is increasing” – include the term “ HIV incidence”. We have made this change.

“In Kenya, of 101 current injectors, 52.5% were found to be HIV positive, attributed mainly to sharing injection equipment compared to 13.5% of heroin users who had not injected” – include reference The reference to this is now included.

“In 2018, the Kampala Place Study estimated over 18,000 people who inject drugs (PWID) in Uganda” – include reference We have included the reference.

“12-step prgrams as MAT was not available at the time of study” – typographical error in programs. “As MAT was not available at the time of study “ should be in parentheses. We have fixed the typo and placed the phrase in parentheses.

Methods:

“Some codes, such as Butabika and drug confiscation by police authorities emerged from the interviews and were added after careful reading of transcripts”. This can be re-written to just say that new themes that emerged were included in the final coding scheme. We have rewritten this sentence as suggested.

Results:

Injection risk behaviors:

“In this situation of scarcity, even people who know the risks of sharing and take precautions to protect themselves reported that they would share their used syringes with other users” – modify tense: In this situation of scarcity, even people who knew the risks of sharing and took precautions to protect themselves reported that they would share their used syringes with other users. We have changed the tense.

“Syringes were also be shared when PWID bought premeasured syringes from drug dealers” – grammatical error. Should be modified as “Syringes were also shared when PWID bought premeasured syringes from drug dealers”. We deleted the “be” from the sentence.

“Unfortunately, the pilot lasted only a year and funding for it had just ended by the time the interviews for this project were conducted, and many participants reported that they had not seen Ronald or received syringes from him for some time”. Ronald is suddenly introduced here. I think this can just be limited to saying, “Unfortunately, the pilot lasted only a year and funding for it had just ended by the time the interviews for this project were conducted”. We have ended the sentence where suggested and eliminated the reference to Ronald.

“Some have heard that HIV is no longer a problem because everyone already has it and there is effective medication”. Change to past tense for consistency with rest of paragraph: Some had heard that HIV was no longer a problem because everyone already had it and there was effective medication. We have changed this to past tense.

“The woman reported that she “can’t” go because she is too afraid and will only go if she is confronted by symptoms”. Again, change to past tense. We changed this to past tense as well.

Sincerely,

Julia Dickson-Gomez

---

## [Editor Report · Decision Letter 3]

6 Apr 2020

Injection and sexual risk among people who use or inject drugs in Kampala, Uganda:   An exploratory qualitative study

PONE-D-19-30396R3

Dear Dr. Dickson-Gomez,

We are pleased to inform you that your manuscript has been judged scientifically suitable for publication and will be formally accepted for publication once it complies with all outstanding technical requirements.

With kind regards,

Joel Msafiri Francis, MD, MS, PhD

Academic Editor

PLOS ONE
---

## [Editor Report · Acceptance letter]

10 Apr 2020

PONE-D-19-30396R3 

Injection and sexual risk among people who use or inject drugs in Kampala, Uganda:   An exploratory qualitative study 

Dear Dr. Dickson-Gomez:

I am pleased to inform you that your manuscript has been deemed suitable for publication in PLOS ONE. Congratulations! Your manuscript is now with our production department. 

With kind regards,

on behalf of

Dr. Joel Msafiri Francis 

Academic Editor

PLOS ONE